# Comparing Inflammatory Biomarkers in Cardiovascular Disease: Insights from the LURIC Study

**DOI:** 10.3390/ijms26157335

**Published:** 2025-07-29

**Authors:** Angela P. Moissl, Graciela E. Delgado, Hubert Scharnagl, Rüdiger Siekmeier, Bernhard K. Krämer, Daniel Duerschmied, Winfried März, Marcus E. Kleber

**Affiliations:** 1Department of Medicine I (Cardiology, Hemostaseology, Medical Intensive Care), Medical Faculty Mannheim, University of Heidelberg, 68167 Mannheim, Germany; angela.moissl@medma.uni-heidelberg.de (A.P.M.); daniel.duerschmied@umm.de (D.D.); 2LURIC Study gGmbH, Josef-Mörtl-Straße 23, 86482 Aystetten, Germany; graciela.delgado@medma.uni-heidelberg.de (G.E.D.); winfried.maerz@synlab.com (W.M.); 3Clinical Institute of Medical and Chemical Laboratory Diagnostics, Medical University of Graz, 8036 Graz, Austria; hubert.scharnagl@medunigraz.at; 4Federal Institute for Drugs and Medical Services, 53175 Bonn, Germany; ruediger.siekmeier@bfarm.de; 5Medical Faculty Mannheim, University of Heidelberg, 68167 Mannheim, Germany; bernhard.kraemer@umm.de; 6Department of Medicine III (Cardiology, Pneumonology, Angiology), Medical Faculty Heidelberg, University of Heidelberg, 69120 Heidelberg, Germany; 7Synlab Academy, SYNLAB Holding Deutschland GmbH, 68161 Mannheim, Germany; 8SYNLAB MVZ Humangenetik Mannheim, 68163 Mannheim, Germany

**Keywords:** serum amyloid A, cardiovascular risk, mortality, high-sensitive C-reactive protein (hsCRP), interleukin-6, inflammation, biomarkers, risk stratification

## Abstract

Inflammatory biomarkers, including high-sensitivity C-reactive protein (hsCRP), serum amyloid A (SAA), and interleukin-6 (IL-6), have been associated with an increased risk of future cardiovascular events. While they provide valuable prognostic information, these associations do not necessarily imply a direct causal role. The combined prognostic utility of these markers, however, remains insufficiently studied. We analysed 3300 well-characterised participants of the Ludwigshafen Risk and Cardiovascular Health (LURIC) study, all of whom underwent coronary angiography. Participants were stratified based on their serum concentrations of hsCRP, SAA, and IL-6. Associations between biomarker combinations and mortality were assessed using multivariate Cox regression and ROC analysis. Individuals with elevated hsCRP and SAA or IL-6 showed higher prevalence rates of coronary artery disease, heart failure, and adverse metabolic traits. These “both high” groups had lower estimated glomerular filtration rate, higher NT-proBNP, and increased HbA1c. Combined elevations of hsCRP and SAA were significantly associated with higher all-cause and cardiovascular mortality in partially adjusted models. However, these associations weakened after adjusting for IL-6. IL-6 alone demonstrated the highest predictive power (AUC: 0.638) and improved risk discrimination when included in multi-marker models. The co-elevation of hsCRP, SAA, and IL-6 identifies a high-risk phenotype characterised by greater cardiometabolic burden and increased mortality. IL-6 may reflect upstream inflammatory activity and could serve as a therapeutic target. Multi-marker inflammatory profiling holds promise for refining cardiovascular risk prediction and advancing personalised prevention strategies.

## 1. Introduction

C-reactive protein (CRP) and serum amyloid A (SAA) are acute-phase proteins primarily synthesised in the liver under the regulation of cytokines, particularly interleukin-6 (IL-6). HsCRP contributes to innate immunity by binding to phosphocholine on microbial surfaces and dying cells, thereby promoting opsonisation and complement activation [1].

Its concentration increases within 6–8 h after an inflammatory stimulus, peaking at 48–72 h, with a relatively stable half-life of approximately 19 h. However, hsCRP is non-specific, as it is elevated in various conditions, including infections, autoimmune diseases, and malignancies. Despite its limitations, hsCRP remains a widely used and reliable marker of systemic inflammation, particularly in cardiovascular risk assessment [2].

SAA comprises a family of apolipoproteins, mainly SAA1 and SAA2, which are also up-regulated in the acute-phase response [3,4]. While their biological roles remain unclear, SAA is involved in immune cell recruitment, lipid metabolism, and modulation of high-density lipoprotein (HDL) composition and function [4,5].

SAA rises rapidly and more sensitively than hsCRP in acute inflammation, sometimes increasing over 1000-fold. Due to its shorter half-life, SAA concentrations fluctuate more dynamically, offering potential advantages in monitoring inflammatory activity; however, its lipid interactions may complicate interpretation [6].

IL-6 is a multifunctional cytokine produced by a wide range of cells, including macrophages, endothelial cells, and T cells, in response to infection, trauma, or stress [7,8].

It acts both locally and systemically, orchestrating the hepatic synthesis of hsCRP and SAA. IL-6 is detectable within 1–2 h of the onset of inflammation and declines rapidly after resolution, making it a sensitive early biomarker for systemic inflammation. It has also emerged as a potential therapeutic target in chronic inflammatory and cardiovascular diseases and [9].

We previously demonstrated, using the Ludwigshafen Risk and Cardiovascular Health (LURIC) study, that SAA was a significant predictor of short-term mortality, whereas IL-6 predicted long-term outcomes [10].

In this study, we extend our investigation by evaluating the combined predictive utility of SAA, hsCRP, and IL-6 in individuals with established or suspected cardiovascular disease.

## 2. Results

### 2.1. Baseline Characteristics According to Inflammatory Biomarker Combinations

Baseline characteristics of the LURIC cohort were assessed based on combined serum concentrations of high-sensitivity C-reactive protein (hsCRP), serum amyloid A (SAA), and interleukin-6 (IL-6), using the respective medians as cut-offs (hsCRP: 3.4 mg/L, SAA: 5.2 mg/L, IL-6: 3.2 ng/L) (Table 1, Table 2 and Table 3).

#### 2.1.1. hsCRP and SAA Stratification

Participants were divided into four groups: those with both biomarkers low (reference), those with high hsCRP only, those with high SAA only, and those with both markers high. Statistically significant differences (*p* < 0.05) were observed across these strata in nearly all clinical and biochemical parameters. Individuals in the “both high” group were older (63.7 ± 10.4 vs. 61.3 ± 10.8 years), more often female (32.6% vs. 25.4%), had higher BMI (28.0 ± 4.52 kg/m^2^ vs. 26.9 ± 3.61 kg/m^2^), and exhibited lower HDL cholesterol (HDL-C) (36.4 ± 10.6 vs. 40.5 ± 10.6 mg/dL) and elevated triglycerides compared to the “both low” group.

Besides hsCRP and SAA, the levels of IL-6 were significantly higher compared to the other groups, consistent with augmented systemic inflammation. Levels of HbA1c and NT-proBNP were highest in the “both high” group, followed by “CRP high” and “SAA high” and lowest in the “both low” group.

For eGFR and albumin, the situation was opposite, with the lowest values in the “both high” group and the highest values in the “both low” group. This indicates adverse metabolic, cardiac, and renal function in patients with higher inflammatory markers.

Cardiovascular disease was more prevalent with increasing inflammation: the percentage of patients suffering from CAD rose from 71.8% in the reference group to 83.6% in the “both high” group (*p* < 0.001). At the same time, heart failure doubled (15.3% to 30.0%). Similarly, diabetes mellitus (32.0% to 47.5%) and smoking, defined as current and former smoking (60.8% to 68.9%), increased.

The Friesinger and Gensini scores, reflecting CAD severity, were significantly elevated in the “both high” group (Table 1).

**Table 1 ijms-26-07335-t001:** Characteristics of LURIC study participants according to median serum hsCRP (<3.4 mg/L or ≥3.4 mg/L) and SAA (<5.2 mg/L or ≥5.2 mg/L) concentration groups ^1^.

Variables	Both Low (N = 1279)	CRP High (N = 372)	SAA High (N = 371)	Both High (N = 1278)	*p*-Value
Age (years)	61.3 ± 10.8	63.2 ± 10.3	63.3 ± 10.4	63.7 ± 10.4	<0.001
Female sex (%)	25.4	28.0	42.9	32.6	<0.001
Body mass index (kg/m^2^)	26.9 ± 3.61	27.9 ± 4.05	27.4 ± 3.81	28 ± 4.52	<0.001
Smoking (current or ex, %)	60.8	70.2	56.3	68.9	<0.001
LDL-cholesterol (mg/dL)	118 ± 33.4	116 ± 33.4	121 ± 37.6	114 ± 34.4	0.008
HDL-cholesterol (mg/dL)	40.5 ± 10.6	36 ± 9.52	43.2 ± 10.8	36.4 ± 10.6	<0.001
Triglycerides (mg/dL)	141 (105–195)	161 (122–222)	140 (106–192)	150 (110–203)	0.006
Systolic blood pressure (mmHg)	141 ± 23.1	144 ± 23.7	143 ± 23.5	139 ± 24.0	0.023
Diastolic blood pressure (mmHg)	81.9 ± 11.1	81.7 ± 11.9	81.9 ± 11.6	79.5 ± 11.5	<0.001
Pulse pressure (mmHg)	59.6 ± 17.7	62.5 ± 18.9	60.7 ± 17.5	59.9 ± 18.6	0.795
hsCRP (mg/L)	1.18 (0.68–1.9)	5.23 (4.22–6.8)	1.82 (1.21–2.61)	10.1 (6.5–22.2)	0
SAA (mg/L)	2.7 (2–3.5)	3.5 (2.7–4.4)	7.1 (5.9–8.85)	16.8 (8.6–43.7)	0
IL-6 (ng/L)	2.01 (1.39–3.32)	3.66 (2.37–5.81)	2.52 (1.58–4.02)	5.62 (3.26–9.99)	<0.001
HbA1c (%)	6.11 ± 1.06	6.45 ± 1.5	6.33 ± 1.19	6.47 ± 1.33	<0.001
Diabetes mellitus (%)	32.0	43.3	37.5	47.5	<0.001
eGFR (mL/min/1.73 m^2^)	86.6 ± 17.7	81.2 ± 19.6	81.8 ± 19	76.9 ± 21.8	<0.001
NT-proBNP (pg/mL)	186 (80–490)	326 (122–854)	267 (109–682)	490 (171–1470)	<0.001
Albumin (g/dL)	4.47 ± 0.56	4.35 ± 0.55	4.43 ± 0.53	4.29 ± 0.54	<0.001
Hypertension (%)	69.7	76.3	74.4	74.3	0.015
Heart failure (%)	15.3	30.3	21.2	30.0	<0.001
Coronary artery disease (%)	71.8	81.2	76.0	83.6	<0.001
Peripheral artery disease (%)	8.76	13.4	4.85	10.8	0.0002
Friesinger score	4 (1–8)	6 (3–9)	5 (2–8)	6 (3–9)	<0.001
Gensini score	24 (1.5–54.5)	34.8 (11.5–68.1)	28.5 (4.5–63)	36.5 (12–72)	<0.001

^1^ Values are presented as either medians (with 25th and 75th percentiles) for non-normally distributed data, means ± standard deviation (SD) for normally distributed data, or percentages for categorical data. *p*-values of less than 0.05 were considered statistically significant. Estimated glomerular filtration rate (eGFR); glycated haemoglobin A1c (HbA1c); high-density lipoprotein (HDL); high-sensitivity C-reactive protein (hsCRP); low-density lipoprotein (LDL); interleukin-6 (IL-6); N-terminal pro-B-type natriuretic peptide-1 (NT-proBNP).

#### 2.1.2. IL-6 and SAA Stratification

Using median cut-offs, participants were classified into IL-6/SAA categories. The “both high” group demonstrated adverse profiles across all domains: older age, female predominance, higher BMI, and more atherogenic lipid patterns. HbA1c and NT-proBNP were significantly elevated. Again, an increase in SAA alone was associated with a smaller increase in metabolic risk markers compared to an increase only in IL-6.

Cardiovascular disease severity paralleled inflammatory status: CAD was observed in 85.7% of the “both high” group, PAD in 11.7%, and HF in 32.2% (all *p* < 0.001). Diabetes mellitus was present in 48.3% (vs. 29.9% in the “both low” group), and smoking prevalence reached 69.0%.

These findings underscore the clinical burden associated with the concurrent elevation of IL-6 and SAA, particularly driven by IL-6, which showed a stronger association with adverse metabolic markers than SAA alone. This supports the utility of combining inflammatory markers to identify a vulnerable cardiometabolic phenotype and highlights potential differential roles of IL-6 and SAA in disease pathogenesis (Table 2).

**Table 2 ijms-26-07335-t002:** Characteristics of LURIC study participants according to median serum IL-6 (<3.2 ng/L or ≥3.2 ng/L) and SAA (<5.2 mg/L or ≥5.2 mg/L) concentration groups ^1^.

Variable	Both Low (N = 1095)	IL6 High (N = 556)	SAA High (N = 551)	Both High (N = 1098)	*p*-Value
Age (years)	60.6 ± 10.8	63.9 ± 10.2	62 ± 10.2	64.4 ± 10.4	<0.001
Female sex (%)	25.5	27	42.8	30.9	<0.001
Body mass index (kg/m^2^)	27 ± 3.77	27.3 ± 3.66	27.8 ± 4.11	27.8 ± 4.5	<0.001
Smoking (current or ex, %)	61.9	64.9	60.1	69	<0.001
LDL-cholesterol (mg/dL)	118 ± 33.6	116 ± 32.9	119 ± 38.6	114 ± 33.3	0.005
HDL-cholesterol (mg/dL)	40.5 ± 10.7	37.5 ± 9.94	40.7 ± 11.3	36.6 ± 10.6	0.001
Triglycerides (mg/dL)	144 (105–198)	150 (113–205)	147 (108–205)	149 (111–199)	0.088
Systolic blood pressure (mmHg)	142 ± 22.7	143 ± 24.4	141 ± 23.1	140 ± 24.3	0.040
Diastolic blood pressure (mmHg)	82 ± 11	81.7 ± 11.8	81 ± 11.1	79.6 ± 11.8	<0.001
Pulse pressure (mmHg)	59.6 ± 17.3	61.5 ± 19.4	60 ± 17.8	60.2 ± 18.7	0.669
hsCRP (mg/L)	1.28 (0.68–2.4)	2.45 (1.21–4.74)	4 (1.73–7.87)	10.3 (5.72–22.8)	<0.001
SAA (mg/L)	2.7 (2–3.6)	3.2 (2.4–4)	8.2 (6.3–14)	16.8 (8.2–45.2)	0
IL-6 (ng/L)	1.73 (1.22–2.29)	5.07 (3.88–7.06)	1.96 (1.42–2.54)	6.98 (4.63–11.6)	0
HbA1c (%)	6.1 ± 1.08	6.36 ± 1.33	6.3 ± 1.07	6.51 ± 1.4	<0.001
Diabetes mellitus (%)	29.9	43.7	39.2	48.3	<0.001
eGFR (mL/min/1.73 m^2^)	87.8 ± 17.5	80.7 ± 18.7	82.5 ± 19.5	75.8 ± 21.8	<0.001
NT-proBNP (pg/mL)	164 (76–426)	333 (129–907)	248 (101–638)	552 (201–1690)	<0.001
Albumin (g/dL)	4.47 ± 0.557	4.4 ± 0.567	4.42 ± 0.498	4.27 ± 0.548	<0.001
Hypertension (%)	69.9	73.7	76.6	73.1	0.029
Heart failure (%)	15.6	24.6	19.7	32.2	<0.001
Coronary artery disease (%)	71.7	78.2	74.4	85.7	<0.001
Peripheral artery disease (%)	8.4	12.6	5.08	11.7	<0.001
Friesinger score	4 (1–7)	6 (2.75–9)	5 (1–8)	6 (3–9)	<0.001
Gensini score	23.5 (1.5–53.5)	32.8 (7–66.1)	26.5 (3–60)	39.2 (17.6–75)	<0.001

^1^ Values are given as either the median (25th and 75th percentiles) for non-normally distributed data, the mean ± SD for normally distributed data, or percentages for categorical data. *p*-values of less than 0.05 were considered statistically significant. Estimated glomerular filtration rate (eGFR); glycated haemoglobin A1c (HbA1c); high-density lipoprotein (HDL); high-sensitivity C-reactive protein (hsCRP); low-density lipoprotein (LDL); interleukin-6 (IL-6); N-terminal pro-B-type natriuretic peptide-1 (NT-proBNP).

##### 2.1.3. hsCRP and IL-6 Stratification

Similarly, stratification by hsCRP and IL-6 confirmed that the “both high” group had the poorest risk profile. Participants were older (64.3 ± 10.4 years), had higher BMI (27.8 ± 4.48 kg/m^2^), and lower HDL-C (35.6 ± 10.1 mg/dL). Signs of metabolic and renal impairment were evident from the highest HbA1c and NT-proBNP concentrations and the lowest eGFR values (76.3 ± 21.6 mL/min/1.73 m^2^).

Clinically, this group showed the highest burden of cardiovascular disease: CAD was present in 85.4%, HF in 32.3%, and PAD in 12.8%, accompanied by elevated Friesinger and Gensini scores. Diabetes mellitus (48.3%) and smoking prevalence (>70%) were also most pronounced, suggesting a clustering of classical and inflammatory risk factors in this high-risk subgroup.

In contrast, the differences observed in groups with only one elevated marker were less marked, particularly when compared to the respective CRP/SAA and IL-6/SAA categories (Table 3).

**Table 3 ijms-26-07335-t003:** Characteristics of LURIC study participants according to serum hsCRP (<3.4 mg/L or ≥3.4 mg/L) and IL-6 (<3.2 ng/L or ≥3.2 ng/L) groups ^1^.

Variable	Both Low (N = 1175)	IL6 High (N = 475)	CRP High (N = 471)	both High (N = 1179)	*p*-Value
Age (years)	60.9 ± 10.8	64 ± 10.4	61.7 ± 10.3	64.3 ± 10.4	<0.001
Female sex (%)	29.3	29.5	36.3	29.6	0.029
Body mass index (kg/m^2^)	26.9 ± 3.69	27.2 ± 3.56	28.3 ± 4.25	27.8 ± 4.48	<0.001
Smoking (current or ex, %)	59.4	60.8	66.0	70.4	<0.001
LDL-cholesterol (mg/dL)	120 ± 35	117 ± 32.8	116 ± 36.2	114 ± 33.4	<0.001
HDL-cholesterol (mg/dL)	41.5 ± 10.8	40.1 ± 10.5	38.2 ± 10.9	35.6 ± 10.1	<0.001
Triglycerides (mg/dL)	140 (103–195)	140 (110–193)	156 (113–218)	151 (114–202)	<0.001
Systolic blood pressure (mmHg)	141 ± 22.6	143 ± 24.6	142 ± 23.3	140 ± 24.2	0.187
Diastolic blood pressure (mmHg)	81.9 ± 10.9	82.1 ± 11.7	81.1 ± 11.3	79.6 ± 11.8	<0.001
Pulse pressure (mmHg)	59.2 ± 17	61.3 ± 19.1	60.9 ± 18.4	60.3 ± 18.8	0.166
hsCRP (mg/L)	1.19 (0.685–1.92)	1.63 (1.05–2.49)	6.1 (4.43–9.02)	9.94 (6.06–21.8)	0
SAA (mg/L)	3 (2.1–4.5)	3.4 (2.4–5.35)	6.8 (4.2–13.9)	13.9 (6.4–41.9)	<0.001
IL-6 (ng/L)	1.68 (1.19–2.24)	4.76 (3.76–6.82)	2.12 (1.58–2.65)	6.82 (4.63–11.2)	0
HbA1c (%)	6.1 ± 1.05	6.33 ± 1.18	6.34 ± 1.16	6.51 ± 1.44	<0.001
Diabetes mellitus (%)	29.4	42.7	42.0	48.3	<0.001
eGFR (mL/min/1.73 m^2^)	87.7 ± 17.3	80.2 ± 18.7	81.8 ± 20.1	76.3 ± 21.6	<0.001
NT-proBNP (pg/mL)	169 (78–415)	320 (120–917)	272 (103–673)	541 (197–1660)	<0.001
Albumin (g/dL)	4.47 ± 0.55	4.43 ± 0.564	4.4 ± 0.505	4.27 ± 0.549	<0.001
Hypertension (%)	69.7	73.3	78.1	73.4	0.005
Heart failure (%)	14.0	23.0	24.6	32.3	<0.001
Coronary artery disease (%)	70.7	77.7	77.3	85.4	<0.001
Peripheral artery disease (%)	7.06	9.89	7.86	12.8	<0.001
Friesinger score	4 (1–7)	6 (2–9)	5 (2–8)	6 (3–9)	<0.001
Gensini score	22.5 (1–53)	32.5 (7–64)	31 (5–61)	38.5 (17–75)	<0.001

^1^ Values are given as either the median (25th and 75th percentiles) for non-normally distributed data, the mean ± SD for normally distributed data, or percentages for categorical data. *p*-values of less than 0.05 were considered statistically significant. Estimated glomerular filtration rate (eGFR); glycated haemoglobin A1c (HbA1c); high-density lipoprotein (HDL); high-sensitivity C-reactive protein (hsCRP); low-density lipoprotein (LDL); interleukin-6 (IL-6); N-terminal pro-B-type natriuretic peptide-1 (NT-proBNP).

### 2.2. Association with Mortality

#### 2.2.1. Combined Impact of Elevated hsCRP and SAA Concentrations on All-Cause and Cardiovascular Mortality

To assess the prognostic relevance of the inflammatory biomarkers, we again stratified the study participants into four groups: those with both biomarkers below median values (reference group), those with high CRP only, those with high SAA only, and those with both high CRP and SAA. Hazard ratios (HRs) with 95% confidence intervals (CIs) were calculated using crude and multivariable-adjusted Cox regression models (Table 4).

After adjustment for conventional cardiovascular risk factors as well as serum albumin and relevant medications (e.g., lipid-lowering therapy, antihypertensive, antiplatelet agents), (Model 2), the “both high” group remained significantly associated with elevated risk of all-cause (HR: 1.25, 95% CI: 1.06–1.48; *p =* 0.009) and cardiovascular mortality (HR: 1.36, 95% CI: 1.10–1.67; *p* = 0.004). However, inclusion of IL-6 in Model 3 attenuated this association, which no longer reached statistical significance (all-cause HR: 1.09, 95% CI: 0.91–1.31; cardiovascular HR: 1.20, 95% CI: 0.96–1.51).

#### 2.2.2. Combined Impact of Elevated Il-6 and SAA Concentrations on All-Cause and Cardiovascular Mortality

To assess the prognostic significance of IL-6 and SAA, participants were categorised into four groups according to median values: both biomarkers below the median (reference group), elevated IL-6 only, elevated SAA only, and both biomarkers elevated. Hazard ratios (HRs) for all-cause and cardiovascular mortality were calculated using Cox regression models, with progressive adjustments for confounding variables (Table 5).

In unadjusted (crude) models, elevated IL-6 alone was associated with significantly higher risks of all-cause (HR: 1.85; 95% CI: 1.52–2.24; *p* < 0.001) and cardiovascular mortality (HR: 1.80; 95% CI: 1.40–2.33; *p* < 0.001). SAA elevation alone was not associated with significant risk. The combination of elevated IL-6 and SAA conferred the highest risk (all-cause: HR: 2.49; 95% CI: 2.12–2.92; *p* < 0.001; cardiovascular: HR: 2.80; 95% CI: 2.28–3.44; *p* < 0.001).

After adjustment for age and sex (Model 1), these associations persisted, particularly in the “both high” group (all-cause: HR: 2.08; 95% CI: 1.77–2.44; *p* < 0.001; cardiovascular: HR: 2.34; 95% CI: 1.90–2.88; *p* < 0.001). Further adjustment for additional cardiovascular risk factors, serum albumin, and relevant medications (Model 2) attenuated the associations, but significant risk remained in the “both high” group (all-cause: HR: 1.50; 95% CI: 1.25–1.81; *p* < 0.001; cardiovascular: HR: 1.70; 95% CI: 1.33–2.17; *p* < 0.001). In the fully adjusted model (Model 3, which additionally included hsCRP), the combination of high IL-6 and SAA remained significantly associated with increased all-cause (HR: 1.35; 95% CI: 1.07–1.70; *p* = 0.010) and cardiovascular mortality (HR: 1.52; 95% CI: 1.14–2.05; *p* = 0.005), whereas isolated SAA elevation was not associated with increased risk in any model.

#### 2.2.3. Combined Impact of Elevated hsCRP and Il-6 Concentrations on All-Cause and Cardiovascular Mortality

To further explore the interplay between systemic inflammation and mortality, participants were stratified by hsCRP and IL-6 concentrations using the same four-group approach. As shown in Table 6, the highest risk was again observed in individuals with both biomarkers elevated.

In crude analyses, both biomarkers were independently associated with significantly increased all-cause and cardiovascular mortality (e.g., IL-6 high: all-cause HR: 1.96; 95% CI: 1.60–2.40; cardiovascular HR: 2.17; 95% CI: 1.67–2.81; both *p* < 0.001). The elevated hsCRP and IL-6 combination showed the strongest associations (all-cause: HR: 2.62; 95% CI: 2.24–3.07; cardiovascular: HR: 2.85; 95% CI: 2.32–3.50; both *p* < 0.001).

Adjustment for age and sex (Model 1) did not materially alter these associations. In Model 2, which included traditional cardiovascular risk factors, serum albumin, and relevant medications, the combination of elevated biomarkers remained significantly associated with all-cause (HR: 1.60; 95% CI: 1.33–1.93; *p* < 0.001) and cardiovascular mortality (HR: 1.69; 95% CI: 1.32–2.15; *p* < 0.001), while hsCRP alone was no longer significant. The fully adjusted Model 3 (additionally accounting for SAA) confirmed these findings, with the “both high” group showing persistent associations (all-cause: HR: 1.62; 95% CI: 1.31–2.00; *p* < 0.001; cardiovascular: HR: 1.53; 95% CI: 1.16–2.01; *p* = 0.002). In this model, isolated IL-6 elevation remained a significant predictor, while hsCRP alone was not significantly associated with mortality outcomes.

We also used clinical cut-offs to define the groups of the inflammatory markers. For hsCRP, values > 3 mg/L were defined as elevated, SAA values > 6.4 mg/L were defined as elevated, and for IL-6, the cut-off was 7 pg/mL. The results using these groups were similar and are presented in Appendix A. The isolated elevation of IL-6 was not significantly associated with mortality in these analyses; however, this may be due to the relatively small number of study participants with values above the threshold (N = 682).

Furthermore, we investigated the association of the group of participants with all three markers above the clinical threshold (N = 1161) with mortality as compared to participant with all markers below the respective threshold (N = 1262) and participants in whom only 1–2 markers were elevated (N = 877). The results are shown in Appendix A. Elevation of all three markers was associated with significantly higher risk as compared to the elevation of only 1–2 markers.

### 2.3. Predictive Performance of Individual and Combined Inflammatory Biomarkers

To compare the prognostic utility of inflammatory markers, receiver operating characteristic (ROC) analyses were performed. The area under the curve (AUC) was used to assess predictive discrimination for each biomarker and their combinations (Figure 1). IL-6 exhibited the highest predictive accuracy for mortality, with an AUC of 0.638. The combination of all three biomarkers achieved an AUC of 0.635, closely matching the prediction from IL-6 alone, suggesting that IL-6 is the dominant contributor to prognostic discrimination in this inflammatory profile.

## 3. Discussion

### 3.1. Inflammatory Biomarker Profiles and Baseline Characteristics

As expected, patients with simultaneous elevations of SAA, hsCRP, and/or IL-6 exhibit a markedly adverse cardiovascular risk profile. These individuals were consistently older, had higher BMI, worse glycaemic control, more unfavourable lipid parameters—characterised by lower HDL-C and higher triglyceride levels—impaired renal function, and a higher prevalence of hypertension, diabetes mellitus, and smoking.

These findings align with previous research that links systemic inflammation to the burden of cardiometabolic diseases [11,12]. Inflammation plays a pivotal role in atherogenesis, and inflammatory markers such as hsCRP and IL-6 have been proposed to serve as key indicators of disease activity [13]. The combination of biomarkers was exceptionally informative: across all stratification models (SAA and hsCRP, SAA and IL-6, and hsCRP and IL-6), the “both high” groups consistently showed the most pronounced abnormalities in metabolic and cardiovascular parameters, as well as significantly higher levels of NT-proBNP.

### 3.2. Prognostic Value of Combined Biomarker Elevation

In our survival analyses, the combined elevation of hsCRP and SAA was significantly associated with both all-cause and cardiovascular mortality in unadjusted and partially adjusted models. This finding supports previous observations on the prognostic importance of inflammatory markers [14].

However, after full adjustment including IL-6, the predictive value of the CRP and SAA combination was attenuated. This aligns with the fact that IL-6 functions as an upstream mediator in the inflammatory cascade, driving the expression of both CRP and SAA [7,9]. Notably, isolated elevations of CRP or SAA were not independently associated with mortality in fully adjusted models including IL-6. A recently published study using data from the LURIC study that examined IL-6 and hsCRP in patients with heart failure also found IL-6 to be the superior marker. It showed a predictive value of elevated IL-6 for CV-mortality in HFpEF but not in HFrEF patients [15]. In line with this observation, IL-6 seemed to be the more informative marker when combined with either SAA or CRP. Conversely, in the fully adjusted models including SAA, only CRP, but not IL-6 was independently associated with mortality.

It is essential to clarify that the term “isolated elevation” in this context refers to participants with one biomarker above and the other below the median, compared to the reference group with both biomarkers below the median. This four-group stratification approach was applied consistently across all analyses and supports the interpretation of IL-6 as the more informative predictor when assessed alongside hsCRP or SAA.

We also investigated risk stratification using a combination of all three markers and could show that these high-risk patients have a significantly higher mortality risk as compared to patients with only 1–2 markers above their clinical threshold. Importantly, these associations remained robust even after comprehensive adjustment for potential confounders, including serum albumin [16,17], which reflects both systemic inflammation and protein status, as well as cardiovascular medications such as lipid-lowering agents, antiplatelet drugs, and beta-blockers. This strengthens the validity of our findings and reduces the likelihood of residual confounding by these factors [14,17,18,19].

### 3.3. Diagnostic Accuracy and Clinical Utility

Receiver operating characteristic (ROC) curve analysis confirmed that IL-6 provided the highest discriminatory power for mortality, with an AUC of 0.638. The combination of all three markers (IL-6, hsCRP, and SAA) yielded an AUC similar to that of IL-6 alone, suggesting that IL-6 contributes most to the prognostic value. These findings underscore the clinical relevance of IL-6 and support its use, alongside hsCRP and SAA, to refine cardiovascular risk stratification. However, as our study is observational, these associations are correlative and do not establish causality. Thus, while our results are consistent with the concept of IL-6 as a potential therapeutic target [14,18], they do not provide direct evidence for this, and further experimental and interventional studies are needed to explore this possibility. Future risk models may nonetheless benefit from incorporating IL-6, particularly in patients with metabolic syndrome, CAD, or residual inflammatory risk despite standard therapies [20].

To further clarify the mechanistic role of IL-6, future studies could explore whether it induces CRP and SAA via JAK-STAT3 signalling in cardiomyocytes [21]—e.g., using hiPSC-derived models—and whether IL-6 inhibition in preclinical models such as high-fat diet rodents can help establish a causal relationship between IL-6 elevation and cardiovascular pathology.

### 3.4. Strengths and Limitations

This study has several limitations. First, its observational design does not allow causal inference. Despite comprehensive multivariable adjustment, residual confounding cannot be entirely excluded. Second, all biomarker measurements were obtained at baseline only, limiting conclusions on longitudinal trends or temporal changes. Third, all participants were of European descent and recruited at a tertiary referral centre, which may restrict the generalisability of our findings to more diverse or population-based cohorts. Fourth, median-based cut-offs were initially used to ensure adequate statistical power for group comparisons. However, we acknowledge that these thresholds may limit clinical interpretability. Therefore, we performed additional analyses using established clinical cut-offs, which yielded broadly similar results and are presented in the Supplementary Material. Among the strengths are the extensive clinical and metabolic phenotyping of participants, standardised coronary angiography data, and long-term follow-up on mortality outcomes. Additionally, the concurrent assessment of SAA, hsCRP, and IL-6 provides a robust framework for evaluating the systemic inflammatory state and its prognostic implications. Collectively, our findings underscore the importance of multi-marker inflammatory profiling in cardiovascular risk assessment and suggest that IL-6 may serve as a key therapeutic and prognostic target in patients with elevated systemic inflammation.

### 3.5. Practical and Analytical Considerations

Although IL-6 demonstrated substantial prognostic and diagnostic value in our study, its routine clinical application remains limited by practical and analytical constraints. Unlike hsCRP and SAA, readily available in clinical laboratories through automated nephelometric or turbidimetric assays, IL-6 measurement is more complex and less accessible in standard care.

In this study, IL-6 was measured using a research-grade enzyme-linked immunosorbent assay (ELISA), which is analytically robust but labour-intensive and not optimised for high-throughput diagnostics. Commercially available chemiluminescence-based platforms, such as the Elecsys IL-6 (Roche Diagnostics, Indianapolis, IN, USA) and Immulite 2000 IL-6 (Siemens Healthineers), offer automated alternatives, but their use remains restricted to specialised laboratories.

Given the strong association of IL-6 with cardiometabolic risk and mortality, broader availability and standardisation of IL-6 assays would facilitate its integration into routine cardiovascular risk stratification. Future studies should evaluate the clinical utility and cost-effectiveness of implementing IL-6 testing in conjunction with established inflammatory markers in diverse healthcare settings.

## 4. Materials and Methods

### 4.1. Study Design and Participants

The analysis included 3300 well-characterised patients undergoing coronary angiography from the LURIC (Ludwigshafen Risk and Cardiovascular Health) cohort [22]. The LURIC study included male and female participants who were referred for coronary angiography due to suspected acute coronary syndromes (ACS), including chest pain and/or non-invasive testing consistent with myocardial ischemia. Exclusion criteria included acute illnesses unrelated to ACS, chronic non-cardiac diseases, malignancies within the past five years, or an inability to provide informed consent.

### 4.2. Clinical Assessment, Definitions, and Endpoints

CAD was defined as the presence of visible lumen narrowing stenosis of ≥20% in at least one of 15 coronary segments according to the American Heart Association (AHA) classification [23]. Increased fasting plasma glucose (≥126 mg/dL) and/or increased glucose after a 2 h glucose tolerance test (>200 mg/dL) and/or increased HbA1c (≥6.5%) and/or a history of diabetes mellitus were used to define diabetes mellitus according to the guidelines of the American Diabetes Association from 2010 [24]. Smoking status was determined using questionnaires and measurements of serum cotinine, using a concentration of 15 µg/L as a threshold for active smoking, as previously described by Delgado et al. [25].

Hypertension was diagnosed if the systolic and/or diastolic blood pressure was ≥140 and/or 90 mmHg, respectively, or if there was a history of hypertension, as per the 2018 ESC/ESH Guidelines for Managing Arterial Hypertension [26]. The estimated glomerular filtration rate (eGFR) was calculated from creatinine and cystatin C using the CKD-EPI eGFR creat-cys formula [27].

Vital status and causes of death were obtained from local registries. Death certificates and hospital records were reviewed independently by two clinicians. The median follow-up was 9.9 years (range: 0.1–11.9 years). Cardiovascular death was defined as sudden cardiac death, fatal myocardial infarction (MI), death from heart failure, procedure-related mortality, fatal stroke, or other CAD-related causes. Nineteen deaths of unknown cause were excluded from cause-specific analyses.

### 4.3. Laboratory Procedures

Fasting blood samples were obtained via venipuncture. Serum lipids were analysed enzymatically on an Olympus analyser using WAKO reagents. Lipoproteins were separated by ultracentrifugation and precipitation, followed by β-quantification.

HsCRP, SAA, and cystatin C were measured by immunonephelometry (N-High-Sensitive CRP; N LATEX SAA; N-Latex Cystatin C, Dade Behring, Marburg, Germany) using a Behring nephelometer II.

Plasma IL-6 levels were measured using an enzyme-linked immunosorbent assay (ELISA) (R&D Systems, Minneapolis, MN, USA). HbA1c was assessed using the UNIMATE 5 immunoassay (Hoffmann-La Roche, Grenzach-Wyhlen, Germany). N-terminal pro-brain natriuretic peptide (NT-proBNP) was measured by electro-chemiluminescence on an Elecsys 2010 (Roche Diagnostics, Indianapolis, IN, USA). Albumin was determined using the ALBUMIN liquicolor assay (Human Gesellschaft für Biochemica & Diagnostica GmbH, Wiesbaden, Germany).

### 4.4. Statistical Analysis

For normally distributed variables, data are presented as means ± standard deviations (SD); for skewed data, medians (along with 25th and 75th percentiles) are reported. Categorical variables are given as percentages. Study participants were categorised into groups according to the median values of hsCRP (3.4 mg/L), SAA (5.2 mg/L), and IL-6 (3.2 ng/L). Group differences were assessed by ANOVA or χ^2^ tests as appropriate. Cox proportional hazard models were built to assess the association with all-cause mortality and cardiovascular mortality.

ROC curves were calculated and compared using the method of Delong as implemented in the R package pROC 1.18.5.

Two-sided *p*-values of less than 0.05 were considered statistically significant. Analyses were performed using R version 4.4.2 and SPSS Statistics version 27.0.0.0 (IBM Corp., Armonk, NY, USA).

## 5. Conclusions

In conclusion, our findings demonstrate that concurrent elevations in SAA, hsCRP, and IL-6 identify a distinct patient subgroup with unfavourable cardiometabolic profiles and increased mortality risk. Among these markers, IL-6 emerged as the strongest individual predictor, likely functioning as an upstream regulator of the inflammatory response. These results advocate for the integration of multi-marker inflammatory profiling into cardiovascular risk models and provide a rationale for targeted anti-inflammatory interventions.

## Figures and Tables

**Figure 1 ijms-26-07335-f001:**
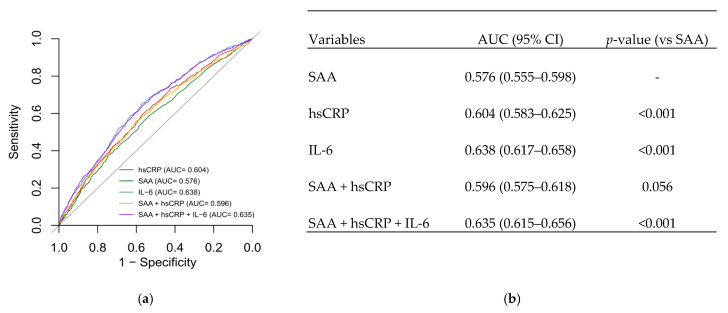
(**a**) ROC curves for SAA, hsCRP, IL-6, and their combinations. (**b**) AUC values and significance compared to SAA alone.

**Table 4 ijms-26-07335-t004:** Association of hsCRP and SAA concentrations with all-cause and cardiovascular mortality ^1^.

	All-Cause Mortality		Cardiovascular Mortality	
	HR (95% CI)	*p*-Value	HR (95% CI)	*p*-Value
Crude				
both low	1	-	1	-
hsCRP high	1.52 (1.23–1.88)	<0.001	1.18 (0.87–1.58)	0.286
SAA high	1.13 (0.89–1.42)	0.319	1.02 (0.75–1.39)	0.910
both high	1.92 (1.66–2.22)	<0.001	2.09 (1.74–2.51)	<0.001
Model 1				
both low	1	-	1	-
hsCRP high	1.38 (1.12–1.71)	0.003	1.07 (0.80–1.44)	0.646
SAA high	1.09 (0.86–1.38)	0.457	0.99 (0.72–1.35)	0.938
both high	1.75 (1.51–2.02)	<0.001	1.90 (1.58–2.29)	<0.001
Model 2				
both low	1	-	1	-
hsCRP high	1.10 (0.87–1.39)	0.447	0.81 (0.58–1.13)	0.214
SAA high	0.92 (0.71–1.18)	0.503	0.83 (0.59–1.16)	0.280
both high	1.25 (1.06–1.48)	0.009	1.36 (1.10–1.67)	0.004
Model 3				
both low	1	-	1	-
hsCRP high	1.02 (0.81–1.29)	0.862	0.76 (0.55–1.07)	0.112
SAA high	0.88 (0.68–1.14)	0.318	0.80 (0.57–1.12)	0.195
both high	1.09 (0.91–1.31)	0.351	1.20 (0.96–1.51)	0.110

^1^ Model 1 was adjusted for age and sex. Model 2 extended this by including body mass index (BMI), low-density lipoprotein cholesterol (LDL-C), high-density lipoprotein cholesterol (HDL-C), albumin, medication (lipid lowering therapy, diuretics, antiplatelet drugs, coumarins, calcium antagonists, AT2 receptor antagonists, cortisol, betablocker), and stratification for the presence of diabetes mellitus, hypertension, smoking status, and estimated glomerular filtration rate (eGFR). Model 3 was Model 2 + (log) IL-6.

**Table 5 ijms-26-07335-t005:** Association of IL-6 and SAA concentrations with all-cause and cardiovascular mortality ^1^.

	All-Cause Mortality		Cardiovascular Mortality	
	HR (95% CI)	*p*-Value	HR (95% CI)	*p*-Value
Crude				
both low	1	-	1	-
Il-6 high	1.85 (1.52–2.24)	<0.001	1.80 (1.40–2.33)	<0.001
SAA high	1.09 (0.87–1.36)	0.437	1.21 (0.92–1.61)	0.176
both high	2.49 (2.12–2.92)	<0.001	2.80 (2.28–3.44)	<0.001
Model 1				
both low	1	-	1	-
IL-6 high	1.53 (1.26–1.86)	<0.001	1.50 (1.16–1.94)	0.002
SAA high	1.09 (0.87–1.36)	0.460	1.21 (0.91–1.61)	0.184
both high	2.08 (1.77–2.44)	<0.001	2.34 (1.90–2.88)	<0.001
Model 2				
both low	1	-	1	-
IL-6 high	1.36 (1.10–1.68)	0.005	1.34 (1.01–1.78)	0.042
SAA high	0.94 (0.74–1.20)	0.622	1.13 (0.83–1.53)	0.438
both high	1.50 (1.25–1.81)	<0.001	1.70 (1.33–2.17)	<0.001
Model 3				
both low	1	-	1	-
IL-6 high	1.33 (1.07–1.65)	0.010	1.31 (0.98–1.74)	0.066
SAA high	0.88 (0.68–1.14)	0.330	1.05 (0.76–1.45)	0.757
both high	1.35 (1.07–1.70)	0.010	1.52 (1.14–2.05)	0.005

^1^ Model 1 was adjusted for age and sex. Model 2 extended this by including body mass index (BMI), low-density lipoprotein cholesterol (LDL-C), high-density lipoprotein cholesterol (HDL-C), albumin, medication (lipid lowering therapy, diuretics, antiplatelet drugs, coumarins, calcium antagonists, AT2 receptor antagonists, cortisol, betablocker), and stratification for the presence of diabetes mellitus, hypertension, smoking status, and estimated glomerular filtration rate (eGFR). Model 3 was Model 2 + (log) hsCRP.

**Table 6 ijms-26-07335-t006:** Association of hsCRP and IL-6 concentrations with all-cause and cardiovascular mortality ^1^.

	All-Cause Mortality		Cardiovascular Mortality	
	HR (95% CI)	*p*-Value	HR (95% CI)	*p*-Value
Crude				
both low	1	-	1	-
IL6 high	1.96 (1.60–2.40)	<0.001	2.17 (1.67–2.81)	<0.001
hsCRP high	1.37 (1.10–1.72)	0.005	1.57 (1.18–2.08)	0.002
both high	2.62 (2.24–3.07)	<0.001	2.85 (2.32–3.50)	<0.001
Model 1				
both low	1	-	1	-
IL6 high	1.64 (1.33–2.01)	<0.001	1.81 (1.39–2.35)	<0.001
hsCRP high	1.36 (1.09–1.70)	0.007	1.55 (1.17–2.06)	0.002
both high	2.18 (1.85–2.55)	<0.001	2.37 (1.92–2.91)	<0.001
Model 2				
both low	1	-	1	-
IL6 high	1.44 (1.15–1.80)	0.001	1.48 (1.11–1.97)	0.007
hsCRP high	1.11 (0.87–1.42)	0.403	1.21 (0.89–1.65)	0.217
both high	1.60 (1.33–1.93)	<0.001	1.69 (1.32–2.15)	<0.001
Model 3				
both low	1	-	1	-
IL6 high	1.44 (1.15–1.80)	0.001	1.47 (1.10–1.96)	0.008
hsCRP high	1.12 (0.87–1.44)	0.388	1.14 (0.83–1.57)	0.420
both high	1.62 (1.31–2.00)	<0.001	1.53 (1.16–2.01)	0.002

^1^ Model 1 was adjusted for age and sex. Model 2 extended this by including body mass index (BMI), low-density lipoprotein cholesterol (LDL-C), high-density lipoprotein cholesterol (HDL-C), albumin, medication (lipid lowering therapy, diuretics, antiplatelet drugs, coumarins, calcium antagonists, AT2 receptor antagonists, cortisol, betablocker), and stratification for the presence of diabetes mellitus, hypertension, smoking status, and estimated glomerular filtration rate (eGFR). Model 3 was Model 2 + log(SAA).

## Data Availability

Due to the German Data Protection Act and participant consent agreements, LURIC study data are not publicly available. However, qualified researchers may request access by contacting Kai Grunwald (kai.grunwald@weitnauer.net) or the Principal Investigator Winfried März (winfried.maerz@luric-online.de or winfried.maerz@synlab.com). Data access will be granted upon approval and completion of a formal agreement process.

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
