# Peer review of "Comparing Inflammatory Biomarkers in Cardiovascular Disease: Insights from the LURIC Study"

_ijms, 2025, doi:10.3390/ijms26157335_

Round 1

Reviewer 1 Report

Comments and Suggestions for Authors

Dr.Kleber and colleagues analyze the prognostic value of three inflammatory biomarkers, hsCRP, SAA, and IL-6 in predicting cardiovascular and all-cause mortality among 3,300 participants from the LURIC cohort who underwent coronary angiography. The authors stratified participants into high and low biomarker groups using the median value for each marker and using multivariate Cox regression and ROC analyses. The results showed that elevated IL-6, alone or in combination with other markers, was most strongly associated with increased risk of both all-cause and cardiovascular mortality, and IL-6 provided the highest independent prognostic value among the biomarkers studied.

I recommend the authors address the following concerns:

  1. Please clarify the rationale for using median-based cut-offs instead of established clinical thresholds, and discuss the limitations of this approach.
  2. Emphasize that the findings are correlative, not causal, and amend the context for suggesting IL-6 as a therapeutic target. The result of this study does not provide concrete evidence for this claim.
  3. I strongly recommend that you conduct a sensitivity analysis using established clinical cut-offs for hsCRP, SAA, and IL-6 in addition to the median-based cut-offs currently used. A sensitivity analysis using these thresholds would demonstrate whether your main findings are robust and consistent, regardless of the cut-off used. Sensitivity analyses ensure that results are not dependent on arbitrary or cohort-specific analytic choices.
  4. If feasible, include a survival analysis for the group with all three biomarkers elevated.

Author Response

Comment: Please clarify the rationale for using median-based cut-offs instead of established clinical thresholds and discuss the limitations of this approach.

Response:

We thank the reviewer for this insightful comment. The rationale for using median-based cut-offs was to ensure sufficiently large groups for the pairwise comparison of inflammatory markers, thereby maintaining adequate statistical power.

In response to the reviewer’s suggestion, we have now performed additional analyses using established clinical cut-offs: hsCRP > 3 mg/L, SAA > 6.4 mg/L, and IL-6 > 7 pg/mL. These results are presented in Supplementary Tables S1–S3 and described in the revised manuscript (page 10, lines 236ff). Overall, the findings were broadly consistent with the main results. However, the group with isolated elevation of IL-6 was relatively small (N = 106 for the comparison of IL-6 and CRP, N = 176 for the comparison of IL-6 and SAA), and isolated IL-6 elevation was no longer significantly associated with mortality after adjustment for risk factors, likely due to limited statistical power.

Furthermore, we examined the association between mortality and the presence of all three markers above their respective clinical thresholds (N = 1161), compared to participants with all markers below the thresholds (N = 1262) and those with only one or two elevated markers (N = 877). These results are presented in Supplementary Figure SF1 and Supplementary Table S4. Notably, participants with all three markers elevated had a significantly higher mortality risk compared to those with one or two elevated markers.

We have also addressed the limitations of using median-based cut-offs in the revised discussion section.

Comment: Emphasise that the findings are correlative, not causal, and amend the context for suggesting IL-6 as a therapeutic target. The result of this study does not provide concrete evidence for this claim.

Response: We appreciate the reviewer’s important observation. As our study is observational in nature, we acknowledge that it is limited to reporting associations rather than establishing causal relationships. We have revised the relevant sections of the manuscript to ensure this distinction is clearly conveyed, particularly with regard to the interpretation of IL-6 as a potential therapeutic target (page 11, lines 307 ff.).

Comment: I strongly recommend conducting a sensitivity analysis using established clinical cut-offs for hsCRP, SAA, and IL-6, in addition to the median-based cut-offs currently used. A sensitivity analysis using these thresholds would demonstrate whether your main findings are robust and consistent, regardless of the cut-off used. Sensitivity analyses ensure that results are not dependent on arbitrary or cohort-specific analytic choices.

Response:

We thank the reviewer for this valuable suggestion. In line with your recommendation, we have performed sensitivity analyses using established clinical cut-offs for hsCRP, SAA, and IL-6, in addition to the median-based cut-offs originally employed. The results of these additional analyses are now included in the Supplementary Material and demonstrate that our main findings remain consistent across different threshold definitions.

Comment: If feasible, include a survival analysis for the group with all three biomarkers elevated.

Response:

We thank the reviewer for this valuable suggestion. In response, we have conducted a survival analysis comparing participants with all three inflammatory biomarkers above the clinical cut-offs, those with only one or two elevated markers, and those with all markers below the thresholds. Kaplan–Meier survival curves illustrating these groups are provided in Supplementary Figure SF1, and the corresponding Cox regression results are included in Supplementary Table S4. This analysis has been described in the revised manuscript (page 11, lines 298–300), where we report that participants with all three elevated markers had a significantly higher mortality risk.

Reviewer 2 Report

Comments and Suggestions for Authors

Dear authors,

I read with great interest your work, which is of great interest to the scientific community, with a good statistical design and discrete results.

In particular, the ROC curves of SAA, hsCRP, IL-6, and their combinations, do not show an AUC 0.635 (0.615-0.656)

<0.001, far from good discriminating power on mortality.

I would probably associate these biomarkers with nutritional data, such as albumin; in fact, in your work, the group with the elevated inflammatory biomarkers has significantly lower serum albumin values than the other groups; in fact, albumin, and in particular the reduced levels, plays a key role in the development of thrombosis.

Therefore, in the various models to obtain HR, I would correct for serum albumin levels. See: doi: 10.1007/s11739-024-03612-9.

Furthermore, in the various models, I would correct for therapies, in particular RAASi, MRAs, SGLT2i, antiplatelets and statins.

Author Response

Comment: I would probably associate these biomarkers with nutritional data, such as albumin; in fact, in your work, the group with the elevated inflammatory biomarkers has significantly lower serum albumin values than the other groups; in fact, albumin, and in particular the reduced levels, plays a key role in the development of thrombosis.

Response:

We thank the reviewer for this insightful comment. While comprehensive nutritional data are not available for our cohort, we agree that serum albumin is an important indicator of nutritional status and thrombosis risk. In our study, participants with elevated inflammatory biomarkers had significantly lower serum albumin levels compared to other groups.

Comment: Therefore, in the various models to obtain HR, I would correct for serum albumin levels. See: doi: 10.1007/s11739-024-03612-9

Response:

We thank the reviewer for this valuable suggestion and for providing the relevant reference. In line with the recommendation, we have included serum albumin levels as an additional covariate in models 2 and 3. The results of these updated analyses are presented in the revised manuscript (page 6, lines 167ff).

Comment: Furthermore, in the various models, I would correct for therapies, in particular RAASi, MRAs, SGLT2i, antiplatelets and statins.

Response:

We thank the reviewer for this important suggestion. In response, we have now included adjustments for relevant medications—including RAAS inhibitors, antiplatelet agents, and statins—in models 2 and 3, as shown in Tables 4, 5, and 6. However, SGLT2 inhibitors were not available at the time of study recruitment, and data on MRAs are unfortunately not available for our cohort, which was enrolled between 1997 and 2000, prior to the widespread clinical use of these agents. This has been clarified in the revised manuscript (page 6, lines 167ff).

Reviewer 3 Report

Comments and Suggestions for Authors

I reviewed this manuscript entitled “Comparing Inflammatory Biomarkers in Cardiovascular Disease: Insights from the LURIC study” by Moissl et al, investigating the combined prognostic utility of three well-known inflammatory markers, including high-sensitivity C-reactive protein (hsCRP), serum amyloid A (SAA), and Interleukin-6 (IL-6) in 3,300 well-characterized participants from the Ludwigshafen Risk and Cardiovascular Health (LURIC) study. All individuals participating in this study have undergone coronary angiography. The authors claimed that the co-elevation of hsCRP, SAA, and IL-6 identifies a high-risk phenotype characterized by a greater cardiometabolic burden and increased mortality. Moreover, authors speculated that IL-6 may reflect upstream inflammatory activity and could serve as a therapeutic target. While the study design is reasonably good, two critical points require the author’s attention to further improve the study’s claim and overall conclusion.

  1. The finding and the conclusion that IL-6 levels alone demonstrated the highest predictive power (AUC: 0.638) and improved risk discrimination when included in multi-marker models are not surprising, but rather confirm numerous studies published in recent past years that have demonstrated the predictive power of IL-6 for cardiovascular mortality. These studies have been reviewed in some of the most recent reviews (https://doi.org/10.3390/jcdd11070206; https://doi.org/10.1007/s11883-024-01259-7; https://doi.org/10.1161/CIRCRESAHA.121.319077). However, it might be exciting to establish a causal link between IL-6 elevation and the emergence of cardiac issues. This can be achieved by conducting a preclinical study in a high-fat diet (HFD) rodent model, where IL-6 can be inhibited using specific inhibitors.
  2. The authors’ reasoning that IL-6 might be working upstream of hcCRP and SAA is indeed supported in hepatic tissue, where it is suggested that IL-6 induces the elevation of hcCRP and SAA through activation of the JAK-STAT3 signaling pathway. Such a pathway should be explored in the context of cardiomyocytes to provide conclusive support for the reasoning. Perhaps, the hiPSC-derived cardiomyocyte model could be harnessed to test modulation of JAK-STAT3 pathway.

Minor points

  1. Table 1 describing the baseline characteristics of the LURIC cohort should also be referenced in the 2.1 section “Baseline characteristics according to inflammatory biomarker combinations” of the results.
  2. The conclusion section should be moved after the discussion.
  3. The prediction of the overall incidence of cardiovascular mortality and/or heart failure by the hsCRP, SAA, either adjusted or unadjusted for IL-6, should be verified using cox proportional hazard model.
  4. An extra letter "s" has been inserted between "augmented and systemic" on line 91. This needs to be removed. 

Author Response

Comment: I reviewed this manuscript entitled “Comparing Inflammatory Biomarkers in Cardiovascular Disease: Insights from the LURIC study” by Moissl et al, investigating the combined prognostic utility of three well-known inflammatory markers, including high-sensitivity C-reactive protein (hsCRP), serum amyloid A (SAA), and Interleukin-6 (IL-6) in 3,300 well-characterised participants from the Ludwigshafen Risk and Cardiovascular Health (LURIC) study. All individuals participating in this study have undergone coronary angiography.

The authors claimed that the co-elevation of hsCRP, SAA, and IL-6 identifies a high-risk phenotype characterised by a greater cardiometabolic burden and increased mortality.

Moreover, authors speculated that IL-6 may reflect upstream inflammatory activity and could serve as a therapeutic target. While the study design is reasonably good, two critical points require the author’s attention to further improve the study’s claim and overall conclusion.

The finding and the conclusion that IL-6 levels alone demonstrated the highest predictive power (AUC: 0.638) and improved risk discrimination when included in multi-marker models are not surprising, but rather confirm numerous studies published in recent past years that have demonstrated the predictive power of IL-6 for cardiovascular mortality. These studies have been reviewed in some of the most recent reviews:

(https://doi.org/10.3390/jcdd11070206; https://doi.org/10.1007/s11883-024-01259-7; https://doi.org/10.1161/CIRCRESAHA.121.319077).

However, it might be exciting to establish a causal link between IL-6 elevation and the emergence of cardiac issues. This can be achieved by conducting a preclinical study in a high-fat diet (HFD) rodent model, where IL-6 can be inhibited using specific inhibitors.

Response:

We thank the reviewer for this thoughtful comment and for providing the relevant references. We fully agree that establishing a causal relationship between elevated IL-6 levels and the development of cardiac disease—such as through preclinical studies using high-fat diet rodent models with IL-6 inhibition—would be of great scientific interest. However, such mechanistic investigations are beyond the scope of our current observational study. We have now highlighted this important point as a promising direction for future research in the revised discussion section (page 12, lines 314–318).

Comment: The authors’ reasoning that IL-6 might be working upstream of hcCRP and SAA is indeed supported in hepatic tissue, where it is suggested that IL-6 induces the elevation of hcCRP and SAA through activation of the JAK-STAT3 signaling pathway. Such a pathway should be explored in the context of cardiomyocytes to provide conclusive support for the reasoning. Perhaps, the hiPSC-derived cardiomyocyte model could be harnessed to test modulation of JAK-STAT3 pathway.

Response:

We thank the reviewer for this insightful comment and fully agree that exploring whether IL-6 induces CRP and SAA via the JAK-STAT3 signalling pathway in cardiomyocytes would provide important mechanistic insight. The suggestion to use hiPSC-derived cardiomyocyte models for such investigations is particularly valuable. While these experimental studies are beyond the scope of our current observational analysis, we have acknowledged this point in the revised discussion as a promising avenue for future research (page 12, lines 314–316), where we state:
To further clarify the mechanistic role of IL-6, future studies could explore whether it induces CRP and SAA via JAK-STAT3 signaling in cardiomyocytes—e.g., using hiPSC-derived models—and whether IL-6 inhibition in preclinical models such as high-fat diet rodents can help establish a causal relationship between IL-6 elevation and cardiovascular pathology.

Minor points

Comment: Table 1 describing the baseline characteristics of the LURIC cohort should also be referenced in the 2.1 section “Baseline characteristics according to inflammatory biomarker combinations” of the results.

Response: We included a reference to the Tables 1-3 in section 2.1.

Comment: The conclusion section should be moved after the discussion.

Response: We appreciate the reviewer's helpful suggestion. As recommended, the conclusion section has been moved to follow the discussion in the revised manuscript.

Comment: The prediction of the overall incidence of cardiovascular mortality and/or heart failure by the hsCRP, SAA, either adjusted or unadjusted for IL-6, should be verified using cox proportional hazard model.

Response:

We thank the reviewer for this suggestion. We confirm that Cox proportional hazard models were used to assess the association of hsCRP and SAA with cardiovascular mortality, both with and without adjustment for IL-6. These results are presented in Tables 4,5, and 6 of the manuscript and for cut-offs in the supplementary material.

Comment: An extra letter "s" has been inserted between "augmented and systemic" on line 91. This needs to be removed.

Response:

We thank the reviewer for pointing this out. The extra letter has been removed from the revised manuscript.

Round 2

Reviewer 2 Report

Comments and Suggestions for Authors

Dear authors,
congratulations on the work, which I found greatly improved. Kudos for the methodology and the new statistical analysis carried out.

Reviewer 3 Report

Comments and Suggestions for Authors

This is an interesting observational study and is expected to benefit the field. No further suggestions are proposed. Acceptance of the MS in its current form is recommended.